# Maslinic Acid Supplementation during the In Vitro Culture Period Ameliorates Early Embryonic Development of Porcine Embryos by Regulating Oxidative Stress

**DOI:** 10.3390/ani13061041

**Published:** 2023-03-13

**Authors:** Ting-Ting Yang, Jia-Jia Qi, Bo-Xing Sun, He-Xuan Qu, Hua-Kai Wei, Hao Sun, Hao Jiang, Jia-Bao Zhang, Shuang Liang

**Affiliations:** Department of Animals Sciences, College of Animal Sciences, Jilin University, Changchun 130062, China

**Keywords:** maslinic acid, porcine early embryos, oxidation resistance, mitochondrial function

## Abstract

**Simple Summary:**

High-quality early embryos are essential for accelerating animal reproduction and genetic modification of mammals, and oxidative stress is strongly associated with a decline in *in vitro* embryo developmental potential. During *in vitro* culture, the most direct and effective approach to alleviate oxidative stress is to add antioxidants to the *in vitro* culture medium. In this study, maslinic acid (MA), a pentacyclic triterpenoid acid in olive plants possessing antioxidant capacities due to its ability to scavenge free radicals, ameliorated the *in vitro* developmental capability of porcine embryos from parthenogenetic activation and somatic cell nuclear transfer. MA also enhanced oxidation resistance, maintained mitochondrial function, and inhibited apoptosis in porcine early-stage embryos.

**Abstract:**

As a pentacyclic triterpene, MA exhibits effective free radical scavenging capabilities. The purpose of this study was to explore the effects of MA on porcine early-stage embryonic development, oxidation resistance and mitochondrial function. Our results showed that 1 μM was the optimal concentration of MA, which resulted in dramatically increased blastocyst formation rates and improvement of blastocyst quality of *in vitro*-derived embryos from parthenogenetic activation (PA) and somatic cell nuclear transfer (SCNT). Further analysis indicated that MA supplementation not only significantly decreased the abundance of intracellular reactive oxygen species (ROS) and dramatically increased the abundance of intracellular reductive glutathione (GSH) in porcine early-stage embryos, but also clearly attenuated mitochondrial dysfunction and inhibited apoptosis. Moreover, Western blotting showed that MA supplementation upregulated OCT4 (*p* < 0.01), SOD1 (*p* < 0.0001) and CAT (*p* < 0.05) protein expression in porcine early-stage embryos. Collectively, our data reveal that MA supplementation exerts helpful effects on porcine early embryo development competence via regulation of oxidative stress (OS) and amelioration of mitochondrial function and that MA may be useful for increasing the *in vitro* production (IVP) efficiency of porcine early-stage embryos.

## 1. Introduction

Compared with *in vivo*-derived embryo collection, IVP of porcine early-stage embryos is a highly essential technique that supplies sufficient quantities of embryos for biotechnology applications such as cloning [1] and genetic modification [2]. Thus, the IVP of high-implantation-potential embryos is an essential step in the process of creating porcine models for biomedical research.

Embryo *in vitro* culture (IVC), a crucial part of the IVP procedure, is a complex system that mimics *in vivo* conditions [3,4]. During IVC, embryos are cultured in controlled laboratory settings with a synthetic medium. Despite substantial progress in the field of IVC optimization, *in vitro*-derived embryos have a lower ratio of early-stage development than *in vivo*-cultured embryos [5]. Long-term accumulation of ROS within embryos during the process of IVC has been established as a significant factor altering *in vitro* embryo quality [6]. Importantly, OS caused by an imbalance between antioxidants and ROS can negatively affect the efficiency of IVC [7]. Excessive ROS exert their pathological effects through damage to cellular organelles [8] and alterations in enzymatic function [9]. Compared with embryos of other species, porcine embryos contain large amounts of stored lipids; therefore, they are more sensitive to environmental conditions in IVC [10]. At present, antioxidant supplementation in the embryo IVC medium is an effective approach to overcome embryonic OS. Various antioxidants, such as resveratrol [11], melatonin [12,13], vitamin C [14], laminarin [15] and asiatic acid [16], have been widely used to alleviate OS by interrupting free-radical chain reactions. However, an optimized system for the effective mitigation of OS-induced embryo damage still needs to be developed.

MA (2α,3β-2,3-dihydroxylolean-12-en-28-oic acid), a plant secondary metabolite, is a triterpenoid compound principally obtained from olive tree (*Olea europaea* L.) [17]; however, it can also be extracted from *Lamprothyrsus hieronymi* Schum. [18], *Tetracentron sinense* Oliv. [19] and *Geum japonicum* [20]. MA and its derivatives exert many beneficial pharmacological effects on cells, such as anticancer [21,22], antidiabetic [23], antiviral [24], antimicrobial [25], anti-inflammatory [26,27,28] and antiplatelet aggregation [29] activities. Extensive studies have suggested that MA can scavenge free radicals as a natural antioxidant [30,31,32,33,34]. Previously, it has been shown that the Akt/Nrf2/HO-1 pathway is involved in the antioxidative activity of MA to vascular smooth muscle cells (VSMCs) [35]. Furthermore, MA-treated diabetic animals have been shown to have reduced malondialdehyde levels, which increases the activity of glutathione peroxidase (GPX) and superoxide dismutase (SOD) in renal, cardiac, and hepatic tissues [36].

Although extensive experiments have shown that MA has a strong ability to scavenge free radicals and relieves oxidative damage to cells or tissues *in vivo* and *in vitro*, the influence of MA on porcine early-stage embryos is rarely reported. Thus, we hypothesized that supplementation with the natural antioxidant MA would be able to relieve systemic OS in porcine early-stage embryos *in vitro*, thereby helping to improve developmental competence. This study first investigated the effects of MA supplementation on the developmental competence of porcine *in vitro*-cultured embryos derived from PA and SCNT. Subsequently, we further analysed the mechanism underlying the promotion of MA on developmental competence of porcine early-stage embryos.

## 2. Materials and Methods

All chemical reagents involved in the study were purchased from Sigma-Aldrich (St. Louis, MO, USA) unless otherwise specified.

### 2.1. Porcine Oocyte Collection and In Vitro Maturation (IVM)

Porcine ovary collection was carried out at a local abattoir within 2 h of obtaining the prepubertal gilts, then delivered to the laboratory in 0.9% saline at 35~37 °C. Cumulus–oocyte complexes (COCs) were collected using a 10 mL syringe aspirating method. According to the IETS grading system, COCs of grade B and above were selected under a stereomicroscope (S22-LGB, Nikon, Shanghai, China), transferred to maturation medium [37] and incubated for 42–44 h at 38.5 °C under 5% CO_2_ in 100% humidified air with mineral oil in 4-well plates (144444, Thermo Fisher, Shanghai, China). Fifty to sixty COCs were cultured in 500 μL of maturation medium per well.

After IVM, the COCs were digested with 0.1% hyaluronidase to remove the surrounding expanded cumulus corona cell. Only the oocytes with homogeneous ooplasm and a polar body were subjected to subsequent experiments.

### 2.2. PA, SCNT and IVC

PA and SCNT procedures were performed according to our previous study [38]. After completing PA and SCNT, approximately 50 activated or reconstructed embryos were transferred to 500 μL porcine IVC medium with mineral oil and continuously cultured in 4-well plates at 38.5 °C, 5% CO_2_ and saturated humidity without changing the medium for 7 days. The day on which the activated or reconstructed embryos were transferred to the IVC medium was denoted as day 0. In this experiment, the percentages of 2-cell and 4-cell embryos, formed blastocysts and hatched blastocysts to the total number of embryos observed on day 2, day 6 and day 7 were counted as cleavage rate, blastocyst formation rate, and hatching rate, respectively.

MA was added to the IVC medium at a final concentration of 1 μM, 2 μM or 5 μM.

### 2.3. Total Cell Number Assay in Blastocysts

To count the total cell numbers in blastocysts, day 6 porcine blastocysts were fixed in 4% paraformaldehyde (*w*/*v*) at room temperature for 30 min. Subsequently, the blastocysts were stained with 10 μg/mL Hoechst 33342 for 10 min. Then, the stained embryos were mounted on glass slides, covered with cover slips and observed under a fluorescence microscope (Axio Vert.A1, Zeiss, Germany). The fluorescence signal intensities in each group of embryos were analysed with ImageJ software (National Institutes of Health, Bethesda, MD, USA).

### 2.4. 5-Ethynyl-2′-Deoxyuridine (EDU) Analysis

The cell proliferation of porcine blastocysts was detected using BeyoClick™ EDU with an Alexa Fluor 555 cell proliferation kit (C0075S, Beyotime, Shanghai, China) as specified in the manufacturer’s protocols. Briefly, blastocysts were incubated with 10 μM EDU at 38.5 °C and 5% CO_2_ in air with saturated humidity for 2 h in the dark. At the end of incubation, the blastocysts were washed with PBS-PVA three times and fixed with 4% paraformaldehyde for 15 min. Then, the blastocysts were permeabilized by treatment with 0.1% Triton X-100 for 10 min and then washed with PBS-PVA three times. Next, the blastocysts were stained with Azide 555 solution for 30 min and 10 μg/mL Hoechst 33342 for 8 min in the dark. After washing with PBS-PVA three times, the blastocysts were mounted on glass slides and observed under a fluorescence microscope. The EDU-positive cells were analysed with NIH ImageJ software.

### 2.5. TUNEL Assay in Blastocysts

In brief, porcine blastocysts were fixed in 4% paraformaldehyde for 1 h. Next, the blastocysts were permeabilized with 0.1% Triton X-100 for 10 min. The blastocysts were washed with PBS-PVA three times and incubated in the dark for 1 h at 37 °C with TUNEL detection solution (MA0223, Dalian Meilun Biotechnology, Dalian, China). Subsequently, the blastocysts were stained with 10 μg/mL Hoechst 33342 for 15 min in the dark. The blastocysts were mounted on glass slides after washing three times and detected under a fluorescence microscope. The apoptotic nuclei in blastocysts were analysed with NIH ImageJ software.

### 2.6. Intracellular ROS and GSH Abundance Analysis

To determine intracellular ROS abundance, porcine 4-cell- or blastocyst-stage embryos were incubated with 10 µM 2′,7′-dichlorodihydrofluorescein diacetate (DCFH-DA; S0033S, Beyotime, Shanghai, China) for 15 min. Furthermore, porcine embryos were cultured in IVC medium with or without 1 µM MA for 2 or 6 days under oxidative damage conditions (200 µM H_2_O_2_ preincubation for 30 min) to define whether MA reversed the oxidative damage in porcine embryos. To determine intracellular GSH levels, porcine 4-cell- or blastocyst-stage embryos were incubated with 10 μM 4-chloromethyl-6,8-difluoro-7-hydroxycoumarin (CMF2HC; C12881, Thermo Fisher, Shanghai, China) for 30 min. The fluorescence signals of both ROS and GSH were captured in tagged image file format (TIFF) using a digital camera connected to the fluorescence microscope, and fluorescence intensities were analysed using NIH ImageJ software.

### 2.7. Western Blot Assay

For Western blotting, embryos in each group (n = 80/per replicate) were collected and roundly lysed in lysis buffer comprising 40% ddH_2_O, 0.5 mM Tris-HCl, 50% glycerol, 10% SDS, bromophenol blue and β-mercaptoethanol at 95 °C. Next, the protein samples were resolved using 12% SDS-polyacrylamide gel electrophoresis (SDS–PAGE), and then transferred to polyvinylidene fluoride (PVDF) membranes. The PVDF membranes were sealed using 5% BSA at room temperature for 2 h; incubated overnight at 4 °C with primary antibodies against OCT4 (1:800, WL03686, Wanleibio, Shenyang, China); CAT (1:4000, 66765-1-Ig, Proteintech, Wuhan, China), SOD1 (1:2000, 10269-1-AP, Proteintech) and GAPDH (1:10,000, 60004-1-Ig, Proteintech); and then incubated with an HRP-conjugated anti-rabbit secondary antibody (1:10,000, SA00001-2, Proteintech) or anti-mouse secondary antibody (1:8000, SA00001-1, Proteintech )for 1.5 h. After washing with 1x TBST three times, the immunoblots were visualized with ECL solution (SQ201, Epizyme, Shanghai, China) by using a Tanon 5200 Image Analyser (Tanon, Shanghai, China) and analysed with NIH ImageJ software.

### 2.8. Reverse Transcription Quantitative Polymerase Chain Reaction (RT–qPCR) Analysis

Day-6 embryos in each group (n = 60/per replicate) were extracted for total RNA using TRIzol Reagent (Takara, Japan). The total RNA was reverse transcribed into cDNA (200 ng) with a Prime Script™ RT Reagent Kit (Takara, Japan). RT–qPCR was performed using SYBR Green Real-Time PCR Master Mix (Roche, Basel, Switzerland) in a PCRmax (Eco, Staffordshire, UK). RT–qPCR was performed in a 10 μL reaction with 5 μL of SYBR Green Real-Time PCR Master Mix, 2 μL of ddH_2_O, 10 μmol in 0.5 μL of forward or reverse primer, and 2 μL of cDNA using the following procedure: 95 °C for 30 s, 95 °C for 5 s and 60 °C for 30 s for 45 cycles. Target gene expression was quantified relative to housekeeping gene (*GAPDH*) expression. Each RT–qPCR primer involved in the process is listed in Appendix A.

### 2.9. Mitochondrial Membrane Potential (ΔΨm) Assay

A ΔΨm assay was carried out using the ΔΨm-sensitive fluorescent probe 5,5′,6,6′-tetrachloro-1,1′,3,3′-tetraethyl-imidacarbocyanine iodide (JC-1; C2003S, Beyotime, Shanghai, China). Briefly, embryos were incubated with 2 μM JC-1 for 30 min. After washing with PBS-PVA three times, the fluorescence signals were captured using a fluorescence microscope connected to a digital camera. The ΔΨm was calculated as the ratio of J-aggregate red fluorescence (red) to J-monomer green fluorescence (green). The fluorescence intensities of each embryo were analysed using NIH ImageJ software.

### 2.10. Intracellular Adenosine 5′-Triphosphate (ATP) Level Analysis

Briefly, embryos in each group (n = 40/per replicate) were lysed with 200 μL of ATP-releasing reagent from an assay kit (S0027, Beyotime, Shanghai, China), sonicated on ice for 10 min and centrifuged at 12,000× *g* at 4 °C for 5 min. Next, ATP assay working solution was prepared by diluting ATP assay reagent with ATP assay diluent at a ratio of 1:4, and then 30 μL of the supernatant was pipetted into a 96-well plate with a pipette together with 100 μL of ATP working solution. Subsequently, chemiluminescence was detected using a microplate reader (Infinite M200 Pro, Tecan, Shanghai, China).

### 2.11. Statistical Analysis

For each experiment, at least three independent biological replicates were required. The data were evaluated using GraphPad 8.0.1 software (GraphPad, San Diego, CA, USA), and statistical comparisons of experiments were made using independent-sample *t* tests. The data are presented as the mean ± standard deviation (SD), and *p* < 0.05 was considered to indicate statistical significance.

## 3. Results

### 3.1. Effect of MA Supplementation at Various Concentrations on Porcine PA Embryo Development

To screen the optimum concentration of MA, parthenogenetically activated embryos were supplemented with different concentrations of MA (control, 1, 2 and 5 μM), and the developmental performance of these embryos was analysed *in vitro*. The results indicated that none of the supplementation experimental groups showed any differences in cleavage rate (Figure 1B; *p* > 0.05). At a concentration of 1 μM, MA prominently increased the blastocyst formation rate on day 6 (Figure 1A,C; 44.97 ± 4.63% vs. 36.70 ± 2.65%; *p* < 0.01) and increased the rate of blastocyst hatching on day 7 (Figure 1D; 12.31 ± 2.38% vs. 8.03 ± 2.09%; *p* < 0.05) for the PA embryos compared with the embryos cultured in the non-supplemented group. Further analyses showed that 1 μM MA not only increased the total cell numbers (Figure 1E,F; *p* < 0.05) and cell proliferation (Figure 2A,B; *p* < 0.05), but also decreased apoptosis occurrence (Figure 2C,D; *p* < 0.01) in these embryos. Given these findings, 1 μM MA was used in all subsequent experiments.

### 3.2. Effect of MA Supplementation on the In Vitro Developmental Potential of SCNT Embryos

Further analyses of the developmental potential of SCNT embryos *in vitro* demonstrated that MA supplementation in IVC prominently improved porcine SCNT embryo developmental competence. Compared with the control, the addition of 1 μM MA to the culture medium did not affect the cleavage rate of SCNT embryos. (Figure 3B; *p* > 0.05). Blastocyst formation rate was clearly higher in the MA-supplemented group than in the control group (Figure 3A,C; 23.15 ± 8.95% vs. 15.12 ± 7.60%; *p* < 0.05). Compared with the non-supplemented group, MA supplementation obviously increased the total cell numbers of blastocysts derived from SCNT embryos (Figure 3D,E; *p* < 0.05).

### 3.3. Effects of MA Supplementation on the Oxidation Resistance of Porcine Early-Stage Embryos

Because MA possesses free radical-scavenging properties, we hypothesized that MA supplementation would increase the resistance of porcine early-stage embryos to OS during the IVC period. The data showed that intracellular ROS abundance in porcine PA embryos was significantly lower at the 4-cell stage (Figure 4A,C; *p* < 0.01) and blastocyst stage (Figure 4B,D; *p* < 0.0001) in the MA supplementation group than in the non-supplemented group. In addition, under oxidative damage conditions, MA supplementation significantly attenuated intracellular ROS levels in porcine parthenogenetic embryos at the 4-cell stage during IVC (Appendix A; *p* < 0.0001) and blastocyst stage (Appendix A; *p* < 0.01) compared with those in the H_2_O_2_-exposed group. Further analysis showed that MA supplementation during IVC increased intracellular GSH abundance in 4-cell embryos (Figure 4E,G; *p* < 0.0001) and blastocyst stage (Figure 4F,H; *p* < 0.01). Western blotting showed that the pluripotency factor octamer-binding transcription factor 4 (OCT4) and antioxidant factors superoxide dismutase 1 (SOD1) and catalase (CAT) were upregulated in the MA supplementation group compared with the non-supplemented group of porcine PA embryos at day 6 (Figure 5). Subsequent RT–qPCR analyses showed that the relative mRNA expression levels of anti-apoptotic gene B-cell lymphoma 2 (*BCL2*), antioxidant-related gene haem oxygenase-1 (*HO-1*) and cell proliferation-related gene dihydroorotate dehydrogenase (*DHODH*) were significantly upregulated, while that of pro-apoptotic gene BCL2-associated X protein (*BAX*) was significantly downregulated, in the MA supplementation group compared with the non-supplemented group of porcine PA embryos at day 6 (Appendix A). These results indicated that MA supplementation could increase the OS resistance of porcine early-stage embryos during the IVC period.

### 3.4. Effects of MA Supplementation during IVC on the Mitochondrial Function of Porcine Early-Stage Embryos

Excess generation of ROS is associated with mitochondrial dysfunction. Thus, the ΔΨm and intracellular ATP levels in porcine parthenogenetic embryos were analysed. The ΔΨm was assayed using the JC-1 fluorescence reaction. Quantitative analysis demonstrated that MA significantly increased the relative ratio of JC-1 fluorescence intensity (red/green) in the MA supplementation group compared with the non-supplemented group of porcine blastocysts derived from PA embryos (Figure 6A,B; *p* < 0.01). Furthermore, analysis indicated that MA supplementation during IVC also led to dramatic increases in intracellular ATP levels (Figure 6C; *p* < 0.0001). These results suggested that MA supplementation could effectively improve the mitochondrial function of porcine early-stage embryos.

## 4. Discussion

During IVC of early-stage embryos, OS often impairs embryo development rate and quality [39]. This paper has indicated evidence to show that MA, a natural antioxidant, promotes the developmental performance of PA and SCNT embryos to blastocysts during the IVC period. This beneficial effect occurred because MA effectively alleviated OS, promoted cell proliferation, reduced apoptosis levels and stabilized mitochondrial function, suggesting that MA can ameliorate the developmental competence of early-stage embryos by enhancing oxidation resistance.

Previous studies have already confirmed that MA has antioxidative effects in several other cell types, including human umbilical vein endothelial cells [27], vascular smooth muscle cells [35], human healthy peripheral blood mononuclear cells [40] and pheochromocytoma cells [31]; however, the present study is the first to evaluate the effect of MA on the *in vitro* development of porcine PA and SCNT embryos during the early stage in IVC with prolonged ROS accumulation. Our data showed that supplementation with 1 µM MA markedly increased the development rate and quality of porcine early-stage embryos, as demonstrated by improved blastocyst formation, hatching and total cell numbers; promoted proliferation; reduced apoptosis; and increased ΔΨm and ATP levels. Importantly, the abundance of intracellular ROS was dramatically decreased and the abundance of intracellular GSH was greatly increased in the MA group. These beneficial effects of MA on the development of early-stage embryos may involve the antioxidant and antiapoptotic properties of MA.

To further explore why MA promotes porcine early-stage embryo development, we used EDU labelling of blastocyst cells, and the results revealed that MA supplementation obviously increased the cell proliferation rate of blastocysts. MA is known to effectively exert pro-proliferative and antiapoptotic influences on somatic cells at a relatively low dose [34]. Similarly, the apoptotic cell numbers of the MA group blastocysts were clearly fewer than the numbers of the non-supplemented group in the present study. *BCL2*-family genes, especially *BAX* and *BCL2*, are key regulators of apoptosis, and their main sites of action are mitochondria [41]. The downregulation of the proapoptotic gene *BAX* and upregulation of the antiapoptotic gene *BCL2* in the MA group also supported our hypothesis [42]. In addition, OCT4 plays a vital role in both cell fate decisions and cell proliferation in porcine early-stage embryos [43], while knockout of OCT4 inhibits blastocyst development [44]. Therefore, the OCT4 expression level is positively related to blastocyst pluripotency. The study also showed that OCT4 was notably upregulated in the MA addition group compared with the non-supplemented group.

Overaccumulation of ROS has adverse effects on cells through damage to cellular lipids [45], DNA and organelles [46], as well as alterations in enzymatic function [47], proliferation and apoptosis [48]. Accordi4ngly, maintaining a dynamic balance between antioxidant and ROS levels is pivotal for high embryo development. GSH, a tripeptide of γ-glutamylcysteinylglycine, is an endogenous antioxidant that contributes to both enzyme-dependent and non-enzymatic dependent against OS [49], and is an ineffective substrate for SOD1-catalysed H_2_O_2_ formation in cells and tissues [50]. A lack of GSH can cause an increase in intracellular ROS generation [51], embryonic apoptosis sensitivity [52] and expended mitochondrial damage [53], and high intracellular GSH levels can improve the developmental competence of embryos by reducing intracellular ROS levels and changing the apoptosis coefficient [54]. In this paper, MA supplementation clearly decreased ROS levels, even under oxidative damage conditions, and evidently increased GSH levels in porcine early-stage embryos, which is consistent with the results of other studies [32]. SOD1 and CAT are crucial enzymatic antioxidants for ROS scavenging [55]; SOD can resolve into a superoxide to oxygen and hydrogen peroxide, which is then converted to water and oxygen by CAT. A prior study reported that MA increases the activity of these enzymes in healthy A10 cells following treatment with H_2_O_2_ [56]. In the present study, SOD1 and CAT levels were markedly higher in the MA group than in the non-supplemented group when porcine zygotes developed to the blastocyst stage. SOD1 and CAT expression were upregulated. Additionally, it has previously been reported that MA protects VSMCs against OS through the *Akt/Nrf2/HO-1* pathway activation [35], and *HO-1* is one of the most critical genes regulated by *Nrf2* [57]. Moreover, it is generally acknowledged that HO-1 deficiency is attributed to embryonic death, and previous studies have confirmed the role of HO-1 in embryonic survival [58]. Consequently, the mRNA expression level of *HO-1* was significantly upregulated, and these results indicated that the antioxidant capacity of blastocysts was greatly improved in the MA group. Hence, MA played a positive role in protecting porcine early-stage embryos from OS by enhancing oxidation resistance.

In addition to scavenging of ROS directly, MA could also maintain mitochondrial stability [59]. Mitochondria are essential organelles during early embryonic development, and mitochondrial dysfunction is associated with failure of embryonic development [60]. ΔΨm is commonly used as an indicator of mitochondrial function and cellular viability in embryos [61]. A normal ΔΨm is necessary for mitochondrial ATP production [62] and oxidative phosphorylation [63]. When ΔΨm decreases, mitochondrial permeability increases [64], ATP synthesis decreases [65], cytochrome c is released [66], intracellular oxidative reduction is altered and *BCL2* gene family members intervene to accelerate apoptosis [67]. Importantly, the release of mitochondrial membrane proteins and the dissipation of ΔΨm occur frequently in disease states with increased cell death [68]. In our study, MA-tre6ated embryos exhibited dramatically increased ΔΨm and ATP levels. In both prokaryotes and eukaryotes, DHODH is the only enzyme in pyrimidine biosynthesis located in the mitochondria rather than the cell membranes [69]; it acts on cell proliferation and has a physical connection with respiratory complexes. Loss of DHODH leads to mitochondrial dysfunction. DHODH deficiency partially inhibited mitochondrial respiratory chain complex III, decreased ΔΨm, and increased ROS production [70]. The upregulation of *DHODH* expression further proved that MA could improve mitochondrial function and cell proliferation in blastocysts.

## 5. Conclusions

The present context suggests that MA can enhance the developmental capacity *in vitro* of porcine early-stage embryos by eliminating OS, enhancing mitochondrial function, promoting proliferation and inhibiting apoptosis, thereby improving the developmental efficiency of early embryos during the IVC period. Therefore, MA is a potential candidate natural antioxidant that can be used to improve the potential of porcine early-stage embryo IVC. In the future, these findings may be verified in *in vitro*-produced porcine embryos and, ultimately, in *in vivo* studies.

## Figures and Tables

**Figure 1 animals-13-01041-f001:**
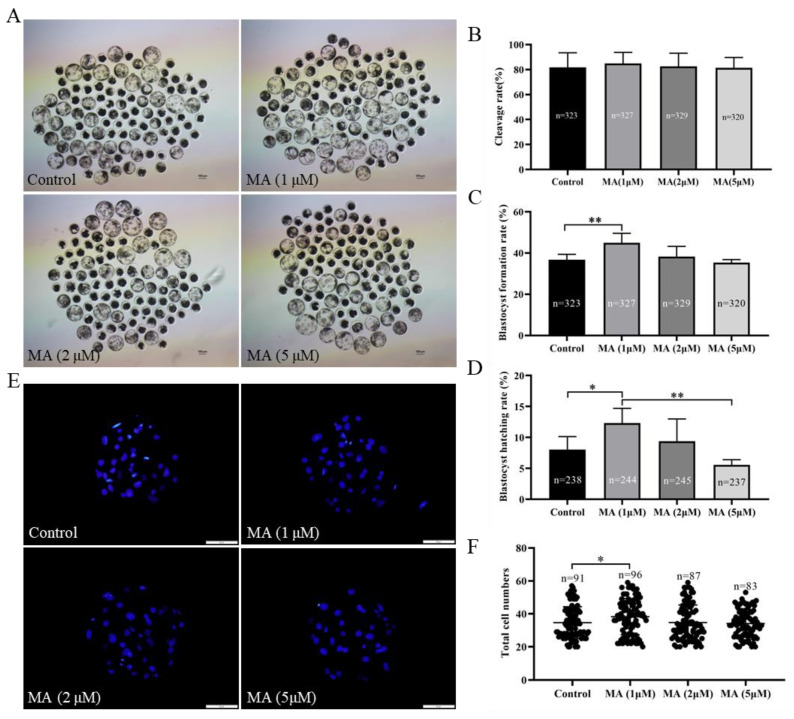
Effect of MA supplementation at various concentrations on the developmental performance of PA embryos. (**A**) Representative images of porcine PA embryonic development on day 6 in the control and MA supplementation groups. Scale bar = 100 μm. (**B**) Cleavage rate of porcine parthenogenetic embryos on day 2. (**C**) Blastocyst formation rate of porcine PA embryos on day 6. (**D**) Blastocyst hatching rate of porcine PA embryos on day 7. (**E**) Representative fluorescence images of blastocyst staining with Hoechst 33342 on day 6 in the control and MA supplementation groups. Scale bar = 50 μm. (**F**) Total cell numbers in blastocysts on day 6. The significant differences are represented with * (*p* < 0.05) and ** (*p* < 0.01).

**Figure 2 animals-13-01041-f002:**
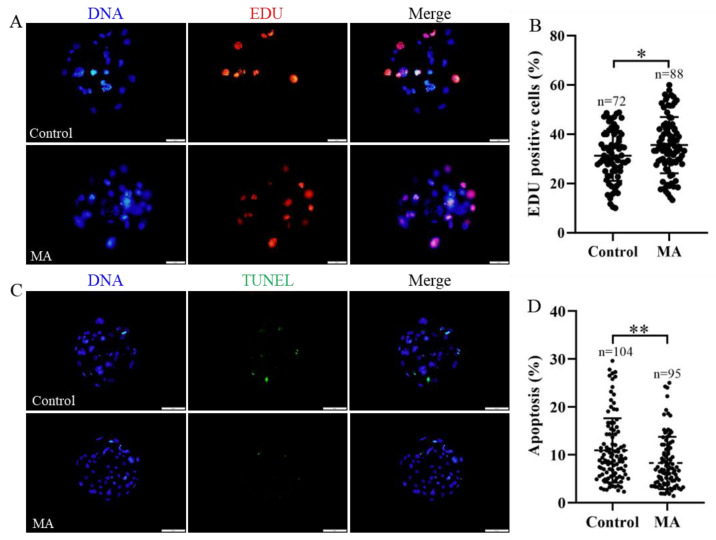
Effect of MA supplementation on cell proliferation and apoptosis in blastocysts derived from porcine parthenogenetic embryos. (**A**) Representative images of EDU-positive cells were detected in blastocysts. Scale bar = 20 μm. (**B**) EDU-positive cell rate in blastocysts on day 6. (**C**) Representative images of TUNEL-positive cells detected in blastocysts. Scale bar = 50 μm. (**D**) Apoptosis rate in blastocysts on day 6. The significant differences are represented with * (*p* < 0.05) and ** (*p* < 0.01).

**Figure 3 animals-13-01041-f003:**
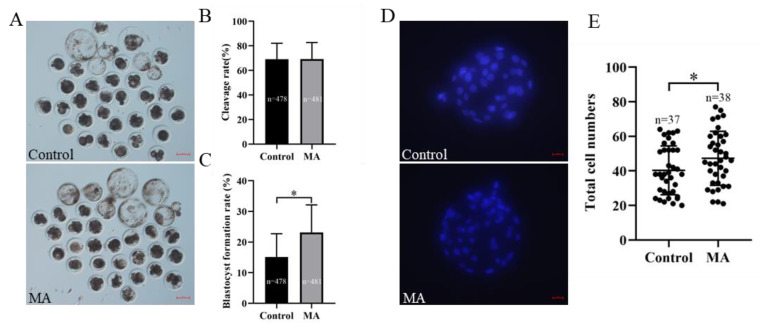
Effect of MA supplementation on the developmental competence of porcine SCNT embryos. (**A**) Representative images of porcine SCNT embryonic development on day 6 in the control and MA supplementation groups. Scale bar = 100 μm. (**B**) Cleavage rate of porcine SCNT embryos on day 2. (**C**) Blastocyst formation rate of porcine SCNT embryos on day 6. (**D**) Representative fluorescence images of blastocyst staining with Hoechst 33342 on day 6 in the control and MA supplementation groups. Scale bar = 20 μm. (**E**) Total cell numbers in blastocysts on day 6. The significant differences are represented with * (*p* < 0.05).

**Figure 4 animals-13-01041-f004:**
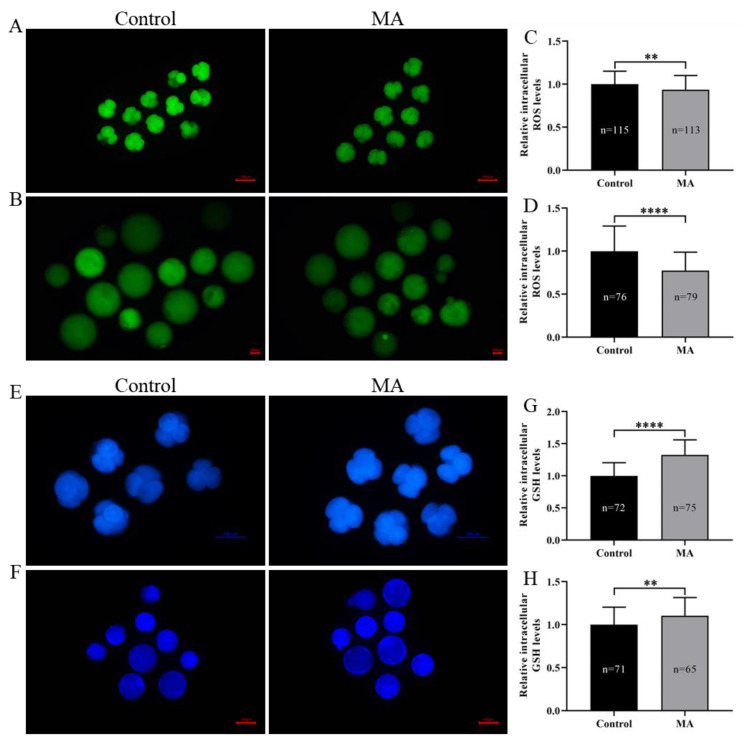
Effect of MA supplementation on antioxidation ability in porcine parthenogenetic embryos. Representative fluorescence images of intracellular ROS in PA embryos at the 4-cell (**A**) and blastocyst (**B**) stages in the control and MA supplementation groups. Scale bar = 100 μm. Relative intracellular ROS levels in 4-cell (**C**) and blastocyst (**D**) stage embryos. Representative fluorescence images of intracellular GSH in parthenogenetic embryos at the 4-cell (**E**) and blastocyst (**F**) stages in the control and MA supplementation groups. Scale bar = 100 μm. Relative intracellular GSH levels in 4-cell (**G**) and blastocyst (**H**) stage embryos. The significant differences are represented with ** (*p* < 0.01) and **** (*p* < 0.0001).

**Figure 5 animals-13-01041-f005:**
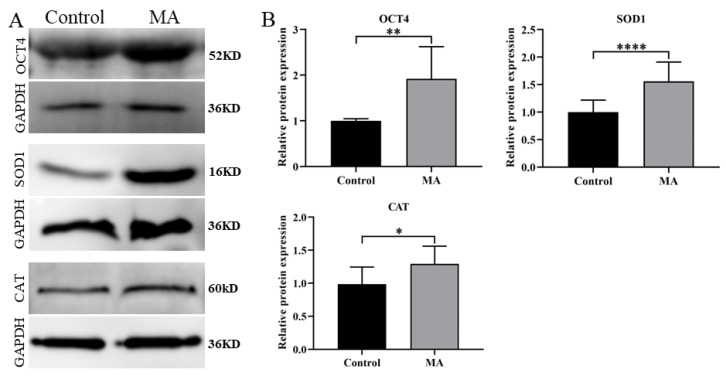
(**A**) Western blotting analysis of OCT4, SOD1 and CAT expression in porcine parthenogenetic embryos in the control and MA supplementation groups. (**B**) Relative expression of OCT4, SOD1 and CAT levels in porcine parthenogenetic embryos. The significant differences are represented with * (*p* < 0.05), ** (*p* < 0.01), **** (*p* < 0.0001).

**Figure 6 animals-13-01041-f006:**
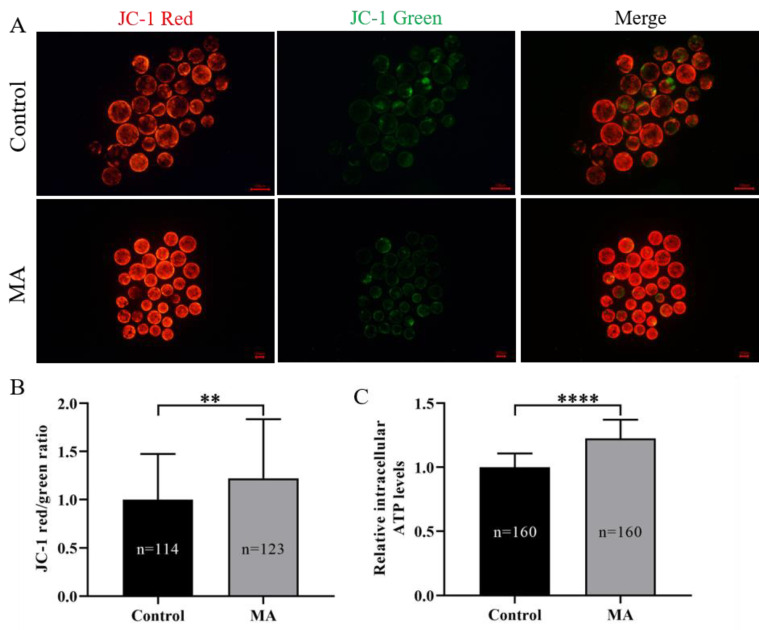
Effect of MA supplementation on ΔΨm and intracellular ATP levels in porcine parthenogenetic embryos. (**A**) Representative fluorescent images of JC-1 staining in porcine PA embryos at the blastocyst stage in the control and MA supplementation groups. Scale bar = 100 μm. (**B**) Relative fluorescence intensity of JC-1 in the blastocyst-stage embryos. (**C**) Relative intracellular ATP levels in blastocyst-stage embryos. The significant differences are represented with ** (*p* < 0.01) and **** (*p* < 0.0001).

## Data Availability

The dataset generated and/or analysed during the current study is available from the corresponding author on reasonable request.

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
