# Peer review of "Maslinic Acid Supplementation during the In Vitro Culture Period Ameliorates Early Embryonic Development of Porcine Embryos by Regulating Oxidative Stress"

_animals, 2023, doi:10.3390/ani13061041_

Round 1

Reviewer 1 Report

The present manuscript "Maslinic acid supplementation during the in vitro culture period ameliorates early embryonic development of porcine embryos derived from parthenogenetic activation and somatic cell nuclear transfer" seemed to be perfectly conceived, designed and performed meticulously. The data was analyzed and presented nicely. 

My only one observation to point out here is that the total cell numbers in the blastocysts shown in figure 1 D are not matching with figure 1E (very less), may kindly check it. 

Otherwise manuscript is worth publication.

Author Response

Thanks for your valuable suggestion. We have revised figure 1 D in our manuscript.

Reviewer 2 Report

Manuscript ID: animals-2240625

Title: Maslinic acid supplementation during the in vitro culture period ameliorates early embryonic development of porcine embryos derived from parthenogenetic activation and somatic cell nuclear transfer

General comments

In this work, the authors test the effect of the antioxidant maslinic acid on the in vitro development of porcine embryos derived from parthenogenetic activation and somatic cell nuclear transfer.

It is an original work. To my best knowledge, the specific antioxidant has not been tested during in vitro culture of porcine embryos. The results add to current knowledge and bear potential merit for the improvement of the efficiency of porcine early-stage embryo production in vitro. Pigs are important farm animals for both agriculture and biomedical research given that they are considered as models for studying human assisted reproduction and may be also used for creating genetically modified animals as potential donors of tissues and organs for xenotransplantation.

The work is correctly designed and implemented. The methodology is presented quite clearly. The results are well presented and interpreted and discussed appropriately. Only a few additions, modifications or clarifications would be suggested (see specific comments below and comments -marked yellow- on the text, in the attached file).

The use of English is good. A few minor suggestions for modifications are marked green on the text, in the attached file.

Specific comments

Line 23: potential mechanisms of What?? Please define.

Line 26: more specific results (i.e., percentages and significant differences where appropriate) should be included in the abstract to give more accurate picture of your findings.

Line 83: empirically; what do you mean? Please be more specific.

Lines 89-102: SUGGEESTION: add references for the composition of the media used.

Line 104: "alive oocytes" in not an accurate description; please characterize oocytes according to IETS grading system.

Line 118 You have chosen the concentrations (molarities) 1 μM, 2 μM or 5 μM, based on what??

Line 119: It seems that did you involve a “negative control” (i.e., medium without DMSO) in your study? Why not??

Lines 120-195 (Materials & Methods): The number of SCNT embryos examined in each of the tests should be mentioned either in paragraphs 2.3., 2.4., 2.5., 2.6. and 2.9. or in the respective paragraphs in the Results section.

Line 150-151: "alive 4-cell- or blastocyst-stage embryos"; “alive” in not an accurate description; please characterize the embryos according to IETS grading system.

Lines 272-273: this is the first time you mention BCL2, HO-1, DHODH, BAX. The connection is not obvious to everybody. Please clarify / add information.

Line 334: “blastocyst activity” this term seems wrong; please replace it or delete it.

Line 404: At the end of the conclusions, I would suggest the addition of a short phrase regarding future research. For example, "In the future, these findings may be verified for in vitro produced porcine embryos and ultimately in in vivo studies."

Figure 2 Correct the legend. In chart (B) you present EDU positive cells and in chart (D) you present apoptosis. In the legend you have written the opposite.

Author Response

Line 23: potential mechanisms of What?? Please define.

Response: Thank you for this helpful suggestion. We have now revised that portion of our manuscript, as shown below.

“The purpose of this study was to explore the effects of MA on porcine early-stage embryonic de-velopment, oxidation resistance and mitochondrial function.”

Line 26: more specific results (i.e., percentages and significant differences where appropriate) should be included in the abstract to give more accurate picture of your findings.

Response: Thank you. We have now added significant differences in the abstract and added percentages of blastocysts for PA and SCNT in our manuscript.

Line 83: empirically; what do you mean? Please be more specific.

Response: Thank you; “empirically” has now been corrected to “further”

Lines 89-102: SUGGEESTION: add references for the composition of the media used.

Response: The composition of the media used has now been referenced.

Line 104: "alive oocytes" in not an accurate description; please characterize oocytes according to IETS grading system.

Response: Thank you very much for drawing this to our attention. We have now corrected the relevant portion of our manuscript.

Line 118 You have chosen the concentrations (molarities) 1 μM, 2 μM or 5 μM, based on what??

Response: The concentrations of MA were chosen based on a preexperiment, and we have revised the relevant portion of our manuscript.

Line 119: It seems that did you involve a “negative control” (i.e., medium without DMSO) in your study? Why not??

Response: We did not set up a negative control group. We compared medium with and without DMSO and found that 0.1% DMSO did not affect porcine embryo embryo development, consistent with a previous study (PubMed: 34407461).

Lines 120-195 (Materials & Methods): The number of SCNT embryos examined in each of the tests should be mentioned either in paragraphs 2.3., 2.4., 2.5., 2.6. and 2.9. or in the respective paragraphs in the Results section.

Response: The numbers of embryos examined in each group are shown in the bars.

Line 150-151: "alive 4-cell- or blastocyst-stage embryos"; “alive” in not an accurate description; please characterize the embryos according to IETS grading system.

Response: We have now corrected the relevant portion of our manuscript.

Lines 272-273: this is the first time you mention BCL2, HO-1, DHODH, BAX. The connection is not obvious to everybody. Please clarify/add information.

Response: We have now added the missing details to our manuscript.

Line 334: “blastocyst activity” this term seems wrong; please replace it or delete it.

Response: Thank you very much for drawing this to our attention. We have revised the relevant portion of our manuscript.

Line 404: At the end of the conclusions, I would suggest the addition of a short phrase regarding future research. For example, "In the future, these findings may be verified for in vitro produced porcine embryos and ultimately in in vivo studies."

Response: Thank you. We have now added “In the future, these findings may be verified for in vitro produced porcine embryos and ultimately in in vivo studies.” at the end of the conclusions.

Point 2: Figure 2 Correct the legend. In chart (B) you present EDU positive cells and in chart (D) you present apoptosis. In the legend you have written the opposite.

Response: Thank you very much for drawing this to our attention. We have revised the relevant portion of our manuscript.

Reviewer 3 Report

In the paper “Maslinic acid supplementation during the in vitro culture period ameliorates early embryonic development of porcine embryos derived from parthenogenetic activation and somatic cell nuclear transfer”, the authors claim to show the beneficial effects of MA on ROS metabolism, embryo development and mitochondrial function. Although this antioxidant has never been used during embryo culture before, I believe some concerns should be addressed before the paper is accepted for publication. 

1.     Some parts of the manuscript are overstated, such as line 40. The context of this statement is regarding embryo development; however, the reference has no connection with embryo development. Many statements are based on review papers, and not on the original papers that present data regarding the conclusions mentioned.

2.     The methodology needs to be improved. For example, when the authors are describing PA and SCNT, an important aspect of the paper, they refer to another paper of the group. In this paper, there is no description of the methodology, they also refer to other papers from the group.

3.     Were the tested doses of MA based on previous studies? How were the tested doses selected?

4.     Western blot membranes, in the original figures, are cut. Because of that, it is hard to evaluate the quality of the western, since it is not possible to check for unspecific bands. 

5.     It is not clear until the results section that SCNT was only used to evaluate embryo development. All the other parameters were evaluated using PA embryos. Cleavage rates are missing. SCNT embryo development is low compared to what is observed by other groups. Based on the pictures, blastocyst rates were calculated over total number of activated/manipulated oocytes. This information should be included. 

6.     The number of embryos used (80 for Western Blott, 60 for RT-qPCR and 40 for ATP/group) was the total number of embryos? The number of embryos per replicate? Please clarify. 

7.     Abstract should include some introduction statement.

8.     The proteins and genes analysed should have been introduced in the introduction or during results presentation, and not only in the conclusion section. 

9.     Figure 6: the presented figure does not represent the difference indicated in the graph. 

10.  Check for abbreviations that are not mentioned before, such as VCMCs (line 72)

11.  Line 55: what are cellular biomacromolecules organelles?

Author Response

Point 1: Some parts of the manuscript are overstated, such as line 40. The context of this statement is regarding embryo development; however, the reference has no connection with embryo development. Many statements are based on review papers, and not on the original papers that present data regarding the conclusions mentioned.

Response: Thank you very much for drawing this to our attention. We have now revised and updated these references in our manuscript.

Point 2: The methodology needs to be improved. For example, when the authors are describing PA and SCNT, an important aspect of the paper, they refer to another paper of the group. In this paper, there is no description of the methodology, they also refer to other papers from the group.

Response: We have now revised and updated these references in our manuscript.     

Point 3: Were the tested doses of MA based on previous studies? How were the tested doses selected?

Response: The concentrations of MA were chosen based on pre-experiment, and we have revised the relevant portion of our manuscript.

Point 4: Western blot membranes, in the original figures, are cut. Because of that, it is hard to evaluate the quality of the western, since it is not possible to check for unspecific bands.

Response: We agree with you. Western blot membranes were washed and incubated with two primary antibodies. The western blot membranes were cut and separated into two membranes. The western blot bands in each original figure are from one PVDF membrane. Our preliminary experiment evaluated the quality of the western blot and found that the western blot membranes did not generate unspecific bands.

Point 5: It is not clear until the results section that SCNT was only used to evaluate embryo development. All the other parameters were evaluated using PA embryos. Cleavage rates are missing. SCNT embryo development is low compared to what is observed by other groups. Based on the pictures, blastocyst rates were calculated over total number of activated/manipulated oocytes. This information should be included.

Response: Thank you for this helpful suggestion. We have now revised that portion of our manuscript

Point 6: The number of embryos used (80 for Western Blott, 60 for RT-qPCR and 40 for ATP/group) was the total number of embryos? The number of embryos per replicate? Please clarify.

Response: We have now revised this portion of our manuscript.

Point 7: Abstract should include some introduction statement.

Response: We have now added an introduction to maslinic acid in the abstract section.

Point 8: The proteins and genes analysed should have been introduced in the introduction or during results presentation, and not only in the conclusion section.

Response: We have now added the missing details to our manuscript.

Point 9: Figure 6: the presented figure does not represent the difference indicated in the graph.

Response: We have now replaced the presented figure in our manuscript.

Point 10: Check for abbreviations that are not mentioned before, such as VCMCs (line 72)

Response: Thank you very much for drawing this to our attention. We have now corrected the relevant portion of our manuscript.

Point 11: Line 55: what are cellular biomacromolecules organelles?

Response: We have now revised the relevant portion of our manuscript.